

**Could the mesoscale eddies be reproduced and predicted in the**
**northern south China sea: case studies**
Dazhi Xu [1], Wei Zhuang[3], Youfang Yan [2*]
[1]South China Sea Marine Prediction Center, State Oceanic Administration,
Guangzhou, China
[2]South China Sea Institute of Oceanology, Chinese Academic of Science, Guangzhou,
China
[3] State Key Laboratory of Marine Environmental Science & College of Ocean and
Earth Sciences, Xiamen University, Xiamen 361102, China
[*]Corresponding author address: Dr. Youfang Yan, State Key Laboratory of Tropical
Oceanography, South China Sea Institute of Oceanology, Chinese Academy of
Sciences, 164 West Xingang Road, Guangzhou, 510301, China.
E-mail: youfangyan@scsio.ac.cn

 

24                                              **Abstract**

Great progress has been made in understanding the mesoscale eddies and their
role on the large-scale structure and circulation of the oceans. However, many
questions still remain to be resolved, especially with regard to the reproduction and
predictability of mesoscale eddies. In this study, the reproduction and predictability of
mesoscale eddies in the northern South China Sea (NSCS), a region with strong eddy
activity, are investigated with a focus on two typical anticyclonic eddies (AE1 and
AE2) based on a HYCOM-EnOI Assimilation System. The comparisons of
assimilated results and observations suggest that generation, evolution and
propagation paths of AE1 and AE2 can be well reproduced and forecasted when their
amplitude >8 cm, although their forcing mechanisms are quite different. However,
when their intensities are less than 8 cm, the generation and decay of these two
mesoscale eddies cannot be well reproduced and predicted by the system. This result
suggests, in addition to dynamical mechanisms, the spatial resolution of assimilation
observation data and numerical models must be taken into account in reproducing and
predicting mesoscale eddies in the NSCS.

**Keywords:** HYCOM; EnOI; Northern South China Sea; Mesoscale eddy;
Predictability





## 1. Introduction

Equivalent to the synoptic variability of the atmosphere, oceanic mesoscale eddies are often described as the "weather" of the ocean, with typical spatial scales of ~100 km and time scales of a month (Chelton et al., 2011; Liu et al., 2001; Wang et al., 1996). The mesoscale eddy is characterized by temperature and salinity anomalies with associated flow anomalies, exhibiting different properties to their surroundings, thus allowing them to control the strength of mean currents and to transport heat, salt, and biogeochemical tracers around the ocean. Although today, the beauty and complexity of these mesoscale features can be seen by viewing high resolution satellite images or numerical model simulations (Yang et al., 2000), the operational forecasts of the mesoscale eddy still poses a big challenge because of its complicated dynamical mechanisms and high nonlinearity (Yuan and Wang, 1986; Li et al., 1998). A recent example is the explosion of the Deepwater Horizon drilling platform in the northern Gulf of Mexico in 2010 where an accurate prediction of the position and propagation of the Loop Current eddy was essential in determining if the spilled oil would be advected to the Atlantic Ocean or still remain within the Gulf (Treguier et al., 2017).

Similar to Gulf of Mexico, the South China Sea (SCS) is also a large semi-closed marginal sea in the northwest Pacific, connecting to the western Pacific mainly through the Luzon Strait (Fig. 1). Forcing by seasonal monsoon winds, the intrusion of Kuroshio Current (KC), the Rossby waves and the complex topography, the SCS, especially the Northern SCS (NSCS) exhibits a significantly high mesoscale eddy



activity (Fig. 2). Many studies have tried to investigate mesoscale eddies in the NSCS
(Wang et al., 2003; Jia et al., 2005; Wang et al., 2008). For instance, based on the
potential vorticity conservation equation and in-situ survey data, Yuan and Wang
(1986) pointed out that the bottom topography forcing might be the primary factor for
the formation of anticyclonic eddies in northeast of Dongsha Islands (DIs). Using
survey CTD data in September 1994, Li et al. (1998) recorded the evidence of
anticyclonic eddies in the NSCS and suggested these anticyclonic eddies are probably
shed from the KC. Using the sea surface height anomaly from satellites, Wang et al.
(2008) found a high frequent occurrence of mesoscale eddies in the NSCS and
indicated that the interaction between strong ocean currents and the local topography
can generate anticyclonic eddies there. Investigations by Wu et al. (2007) showed that
westward propagating eddies in the NSCS originate near the Luzon Strait rather than
coming from the western Pacific. These studies improved our understanding of
activities of mesoscale eddy and its possible dynamical mechanisms in the NSCS.

Although the occurrence and possible dynamical mechanisms of mesoscale eddies

in the NSCS have received much attention in past decades, studies on the
reproduction and predictability of mesoscale eddies in the NSCS are still rare. Since
mesoscale eddies are related not only to complicated dynamical mechanisms but also
involve strong nonlinear processes (Oey et al., 2005), thus they are not a deterministic
response to atmospheric forcing. The quality of mesoscale eddies forecasting will
depend primarily on the quality of the initial conditions. Ocean data assimilation,
which combines observations with the numerical model, can provide more realistic



initial conditions and thus is essential for the prediction of mesoscale eddies. In this
study, we assessed the reproduction and predictability of two typical anticyclonic
eddies in the NSCS with focus on their generation, evolution and decay processes by
a series of numerical experiments based on a Chinese Shelf/Coastal Seas Assimilation
System (CSCASS; Li, 2009; Li et al., 2010; Zhu, 2011), along with the observations
from surface drifter trajectory and satellite remote sensing.
**2. Datasets and Methodologies**
**2.1 Datasets**
In this study, the altimetric data between 2003-2004, which includes along-track
SLA, totally 29 passes (about 9300 points) over the NSCS was selected. Considering
the noise of SLA measurement in the shallow seas, data for the shallow areas with
depth<400 m was excluded. In order to verify assimilation results, the merged SLA
based on Jason-1, TOPEX/Poseidon, ERS-2 and ENVISAT (Ducet et al., 2000)
provided by Archiving, Validation and Interpretation of Satellites Oceanographic data
(AVISO)      at      Centre      Localization      Satellite      (CLS,
ftp://ftp.aviso.oceanobs.com/global/nrt/)  with  1/4° x 1/4°  resolution  and  weekly
average are used. In addition, because the SLA present only the anomalies relative to
a time-mean sea level field, a new mean dynamic topography (nMDT), which has
been corrected using iterative method by Xu et al. (2012) was used to calculate the
realistic sea level in this study.
In addition to SLA datasets, we also used the daily OISST from the National



Oceanic and Atmospheric Administration's (NOAA) National Climatic Data Center
(ftp://eclipse.ncdc.noaa.gov/pub/OI-daily-v2/NetCDF/), which was merged by an
optimum interpolation method (Reynolds et al., 2007) based on the Infrared SST
collected by the Advanced Very High Resolution Radiometer sensors on the NOAA
Polar Orbiting Environmental Satellite and SST from Advanced Microwave Scanning
Radiometer for the Earth Observing System. The daily OISST's biases were fixed
using in situ data from ships and buoys. The dataset between 2003 and 2004 was used
in this study, with a spatial resolution of 1/4°×1/4°. In addition, the surface drifting
buoy    data    from    the    World    Ocean    Circulation    Experiment    (WOCE,
ftp://ftp.aoml.noaa.gov/pub/phod/buoydata/) are also used. A total of 3 drifters
designed to drift at the surface within the upper 15 m were tracked by the ARGOS
satellite system. Positions of the drifters were smoothed using a Gaussian-filter scale
of 24 h to eliminate tidal and inertial currents, and were subsampled at 6-h intervals
(Hamilton et al., 1999).
**2.2 Method to identify the mesoscale eddies**

Similar to the standard of Chelton et al., (2011) and Cheng et al., (2005), we

identify the mesoscale eddies in this study as follows: 1) there must be a closed
contour on the merged SLA; 2) there must be one maximum or minimum inside the
area of closure contour for anticyclonic or cyclonic eddy; 3) the difference between
the extremum and the outermost closure of SLA, that is, the intensity of the mesoscale
eddy must be greater than 2 cm; and 4) the spatial scale of the eddy should be 45-500
km. In addition, the amplitude (A) of an eddy is defined here to be the magnitude of




the difference between the estimated basal height of the eddy boundary and the
extremum value of SSH within the eddy interior: A=|$h_{ext}$-$h_0$|.

**2.3 Ocean model**

We here used a three-dimensional hybrid coordinate ocean model (HYCOM;
Bleck, 2002; Halliwell et al., 1998; 2000; Halliwell, 2004; Chassignet et al., 2007) to
provide a dynamical interpolator of observation data in the assimilation system.
HYCOM is a primitive equation general ocean circulation model with vertical
coordinates: isopycnic coordinate in the open stratified ocean, the geopotential (or z)
coordinate in the weak stratified upper ocean, and the terrain following
sigma-coordinate in shallow coastal regions. The general equations and numerical
algorithms of model in terms of three dimensions velocity field $\vec{u}(u,v,w)$, pressure $p$,
in situ density $\rho$ and the conservation of temperature ($\theta$) and salinity ($S$) are
follows:

$$\frac{\partial}{\partial t_s}\left(\frac{\partial p}{\partial s}\right) + \nabla_s \bullet \left(\bar{V}\frac{\partial p}{\partial s}\right) + \frac{\partial}{\partial s}\left(\frac{\partial s}{\partial t}\frac{\partial p}{\partial s}\right) = 0 \tag{1}$$

$$\frac{\partial \bar{V}}{\partial t_s} + \nabla_s \frac{\bar{V}^2}{2} + \left(\xi + f\right)\vec{k}\times\vec{V} + \left(\frac{\partial s}{\partial t}\frac{\partial p}{\partial s}\right)\frac{\partial \bar{V}}{\partial p} + \nabla_s M - p\nabla_s\alpha = \tag{2}$$
$$-g\frac{\partial \bar{\tau}}{\partial p} + \left(\frac{\partial p}{\partial s}\right)^{-1}\nabla_s \bullet \left(\vartheta\frac{\partial p}{\partial s}\nabla_s\bar{V}\right)$$

$$\frac{\partial}{\partial t_s}\left(\frac{\partial p}{\partial s}\theta\right) + \nabla_s \bullet \left(\bar{V}\frac{\partial p}{\partial s}\theta\right) + \frac{\partial}{\partial s}\left(\frac{\partial s}{\partial t}\frac{\partial p}{\partial s}\theta\right) =$$
$$\nabla_s \bullet \left(\vartheta\frac{\partial p}{\partial s}\nabla_s\theta\right) + \hbar_\theta \tag{3}$$

where $p$ is pressure,$s$ is the vertical coordinate,$\vec{V}=(u,v)$ is the horizontal velocity,



$\xi = \partial v / \partial x_s - \partial u / \partial y_s$ is relative vorticity, $M = gz + p\alpha$ is Montgomery function,
$\theta = gz$ is the gravitational potential,$\alpha$ is the specific volume; $f$ is the Coriolis
parameter,$\vec{k}$ is the unit vector in the vertical direction,$\vartheta$ is viscosity coefficient,
$\tau$ is the wind stress.
In this study, HYCOM was implemented in the Chinese shelf/coastal seas with a
horizontal resolution of $1/12° \times 1/12°$, and in the remaining regions with $1/8° \times 1/8°$, the
model domain is from 0°N to 53°N and from 99°E to 143°E, the detail model domain
and grid are shown in the inset panel of Fig.1. The vertical water column from the sea
surface to the bottom was divided into 22 levels. The K-Profile Parameterization
(KPP; Large et al., 1994), which has proved to be an efficient mixing
parameterization in many oceanic circulation models, was used here. The bathymetry
data of the model domain were taken from the 2-Minute Gridded Global Relief Data
(ETOPO2).
To adjust the model dynamics and achieve a perpetually repeating seasonal cycle
before applying the interannual atmospheric forcing, the model was initialized with
climatological temperature and salinity from the World Ocean Atlas 2001 (WOA01;
Boyer et al., 2005) and was driven by the Comprehensive Ocean-Atmosphere Data
Set (COADS; Woodruff et al., 1987) in the spin-up stage. After integrating ten model
years with climatological forcing, the model was forced by the European Center for
Medium-Range Weather Forecasts (ECMWF) 6-hourly reanalysis dataset (Uppala et
al., 2005) from 1997 to 2003. The wind velocity (10-m) components were converted
to stresses using a stability dependent drag coefficient from Kara et al. (2002).



Thermal forcing included air temperature, relative humidity and radiation (shortwave
and longwave) fluxes. Precipitation was also used as a surface forcing from Legates et
al. (1990). Surface latent and sensible heat fluxes were calculated using bulk formulae
(Han, 1984). Monthly river runoff was parameterized as a surface precipitation flux in
the ECS, the SCS and Luzon Strait (LS) from the river discharge stations of the
Global Runoff Data Centre (GRDC) (http://www.bafg.de), and scaled as in Dai et al.
(2002). Temperature, salinity and currents at the open boundaries were provided by an
India-Pacific domain HYCOM simulation at 1/4° spatial resolution (Yan et al., 2007).
Surface temperature and salinity were relaxed to climate on a time scale of 100 days.
Both two-dimensional barotropic fields such as Sea Surface Height and barotropic
velocities, and three-dimensional baroclinic fields such as currents, temperature,
salinity and density were stored daily.
**2.4 The assimilation scheme**

The ensemble optimal interpolation scheme (EnOI; Oke et al., 2002), which is

regarded as a simplified implementation of the EnKF, aims at alleviating the
computational burden of the EnKF by using stationary ensembles to propagate the
observed information to the model space. The data assimilation schemes can be
briefly written as (Oke et al., 2010):
$$\bar{\psi}^a = \bar{\psi}^b + K(\bar{d} - H\bar{\psi}^b)$$    (4)
$$K = P^b H^T [H P^b H^T + R]^{-1}$$    (5)
where $\bar{\psi}$ is the model state vectors including model temperature, layer thickness and





velocity; Superscripts $a$ and $b$ denote analysis and background, respectively; $\bar{d}$ is
the measurement vector that consists of SST and SLA observations; $K$ is the gain
matrix; and $H$ is the measurement operator that transforms the model state to
observation space; $R$ is the measurement error covariance. In EnOI, Eq. 5 can be
expressed as:
$$K = \varphi(\sigma \circ P^b)H^T[\varphi H(\sigma \circ P^b)H^T + R]^{-1} \qquad (6)$$
where $j$ is a scalar that can tune the magnitude of the analysis increment; $\sigma$ is a
correlation function for localization; $P^b$ is the background error covariance, which
can be estimated by
$$P^b = A'A'^T/(n-1) \qquad (7)$$
In Eq. 7, $n$ is the ensemble size, $A'$ is the anomaly of the ensemble matrix,
$A = (\psi_1, \psi_2, \cdots, \psi_N) \in \mathfrak{R}^{n \times N}$ $(\psi_i \in \mathfrak{R}^N (i = 1, \cdots, n)$ is the ensemble members, $N$ is the
dimension of the model state, representing usually the model variability at certain
scales by using a long-term model run or spin-up run. More detailed description and
evaluation of the CSCASS are in Li et al., (2010) and Xu et al., (2012).

**3. Results**
**3.1 Observations of two anticyclonic eddies in the NSCS**
In this study, we investigated two representative anticyclonic eddies in the NSCS,
one generated in the interior (named AE1) and another shed from the Kuroshio loop
(named AE2). The AE1 generated by interaction of the unstable rotating fluid with the
sharp topography of DIs (Wang et al., 2008) firstly appeared near DIs on the 10[th] of



December 2003 (see Fig. 3). Then it began to move southwestward with its amplitude
decreasing gradually. During the movement of AE1, another anticyclonic eddy (AE2)
was shed and developed from the loop current of Kuroshio near the Luzon Strait. The
amplitude of AE2 was then increased when it propagated southwestward (Fig. 3d-3f).
About five weeks later, AE2 reached its maximum in amplitude and then lasted
around three weeks in its mature state. During its decay phase, AE2 moved
southwestward quickly with its amplitude decreasing, and finally disappeared at the
location of 114°E, 18°N. In the meanwhile, AE1 continued moving to southwest and
eventually disappeared in southeastern of Hainan.
**3.2 The reproduction of these anticyclonic eddies in the NSCS**
In order to investigate whether the evolution and migration features of these two
eddies can be reproduced by the CSCASS or not, we firstly set up an assimilation
experiment named As_exp (see Table 1) for AE1 and AE2. In this experiment, the
observed SST and SLA are both assimilated into CSCASS at 3 days interval. To
enable dynamic adjustment, the first assimilation was performed on the $27^{th}$ of
September 2003, two months prior to the generation of AE1. Figure 4 compares the
assimilating results of AE1 with the satellite remote sensing and trajectories of drifter
buoys number 22517, 22918 and 22610 between December $3^{rd}$ 2003 and February
$18^{th}$ 2004. From Fig. 4 and Table 2, we can see that the generation and movement of
AE1 can be well reproduced by the CSCASS, with the pink curves (assimilation)
match well with those of black (satellite observations) and dotted lines    (the
trajectories of drifter buoys). In addition, the spatial pattern of AE1 can also be well



revealed by the CSCASS: the meridional and zonal radii of AE1 detected by the
assimilation are 163 km and 93 km, which are almost equal to that of observations
(148 km and 79 km). The migration path of AE1 can also be well reproduced by the
CSCASS (see Fig. 4, black and pink line) until its amplitude decays to less than 8 cm.
In addition to AE1, the generation and evolution of AE2 are also evaluated. As shown
in Fig. 5, the evolution and propagation pathway of AE2 (Fig. 5b-5j), e.g., moving
firstly northwestward and then southwestward, can generally be reproduced by the
CSCASS, although its initial location shows a slight southward bias in the simulation
(Fig. 5a). Similar to the results of AE1, discrepancies between model and observations
become larger again during the decay phase of AE2.

In general, the comparison of assimilation SLA with that of satellite observation

and the trajectories of drifter buoys suggested that the generation, development and
the propagation of AE1 and AE2 can be reproduced by the CSCASS when their
amplitude greater than 8 cm. However, when their intensity is relatively weak, with
amplitudes less than 8 cm, the features of these two mesoscale eddies are not well
reproduced by the CSCASS. This may be related to the value setting of parameter α,
the localization length scale, and insufficient spatial resolution of assimilating SSH or
the numerical model (Counillon and Bertino, 2009).
**3.3 The predictability of these anticyclonic eddies in the NSCS**

Since the generation, development and the propagation of AE1 and AE2 can be

well reproduced by the CSCASS when their amplitude>8 cm, as mentioned above, in
this section we further use the CSCASS to investigate the predictability of these two



eddies. According to the generation, evolution and migration of these two eddies, we
designed six forecast experiments, hereafter referred to as Exp1 to Exp6 (see Table 1)
to investigate their predictability. The model's initial state prior to each of the six
forecast experiments is constrained by assimilating satellite SLA and SST beforehand.
Based on the initial state, each experiment is run forward 30 days with the forcing of
6-hourly wind, surface heat flux, and monthly mean river runoff, etc. The first
experiment, named Exp1, is applied on the 29th of November 2003, which tends to
study whether the generation of AE1 can be forecasted or not. Exp2 is implemented
on the 10th of December 2003 and is used to study whether the development and the
migration of AE1 can be forecasted. Exp3 is run based on the initial state on the 31th
of December 2003 and used to show whether the generation of AE2 and the continued
migration of AE1 can be forecasted. In order to investigate whether the continued
evolution of AE1 and AE2 can be forecasted, Exp4 is applied on the 21th of January
2003. Exp5 is set up to reveal whether the attenuation of AE1 and the evolution of
AE2 can be forecasted, while Exp6 which is applied on the 29th of February 2004 was
designed to find out whether the disappearance of AE1 and AE2 can be forecasted.

The prediction results of Exp1 are shown in Fig. 6. In Fig. 6a, we can see that the

forecast is almost coincident with the satellite observation and the trajectory of drift
buoys, indicating that the generated position of AE1 can be well forecasted by the
CSCASS. In addition, the initial migration of AE1 can also be forecasted by the
CSCASS (see Fig. 6a and 6f). In order to evaluate the forecasted amplitude of AE1,
the intensity, amplitudes of eddy centers between the observation and the forecast are



also quantified (Table 3: EXP1). From Table 3: EXP1, we can see that the amplitude
of forecasting matches well with that of observation, although its amplitude is slightly
larger than that of observation. After 4 weeks, the amplitude and intensity of the
forecast are still close to those of the observation, suggesting that the generation of
AE1 can be well predicted by the CSCASS.

In order to find out whether the development and movement path of AE1 can be

predicted after generation, we continue to carry out Exp2. As shown by the
observation (Fig. 7), AE1 moves southwestward along the continental shelf with its
amplitude decreasing and again increasing after its generation. This observed
southwestward movement is also predicted by the CSCASS (see pink closure curve in
Fig. 7a-7d), although a sudden southwestward movement cannot be well predicted
(Fig. 7f). In addition, the first attenuation and then enhancement of AE1 is also
predicted by the CSCASS (see Table 3 and Fig. 7b). On the whole, the development
and movement path of AE1 can be well predicted by CSCASS for the first four weeks
after its generation. After that, the errors between observation and prediction increase
significantly, and by the fifth week, the distance between the center of the prediction
and the observation become larger, more than 100 km (see Fig. 7e).

For further analysis, we carry out Exp3, to look at whether the continued

evolution of AE1 and the generation of AE2 can be predicted. This experiment is
carried out based on the initial condition of the assimilation on the 31$^{st}$ of December
2003. The development trend of AE1 can be predicted, but with a slightly weak
amplitude, as shown by the prediction (Fig. 8, Table 3). The observed center elevation

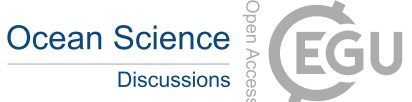

of AE1 reduced from 18 cm in the first week to 13 cm in the fifth week. Similar trend
was also found for the forecast but with its amplitude decreasing from 13 cm at the
beginning to 10 cm at the end of the forecast period. Although the decreasing trend of
AE1 amplitude is quite similar between the observations and forecast, their intensity
is slightly different. In addition, the movement path of AE1 cannot be accurately
predicted at this period, for instance, the observed AE1 moves directly to southwest
(see red solid line and solid circle in Fig. 8f), but the prediction's movement is firstly
toward northeast, then turns to southwest (see blue solid line and solid circle in Fig.
8f). The generation of AE2 cannot be predicted in Exp3, which may be related to the
lower amplitude (<8 cm) of AE2 at this period.

The purpose of Exp4 is to look at whether the evolution of AE1 and AE2 can

both be reasonably predicted. Since this experiment mainly focuses on the evolution
of AE2, thus Fig. 9 shows only the evolution of AE2 from the second week after
generation, that is, from the beginning on the 21$^{st}$ of January 2004 to the fifth week.
As shown in Fig. 9, Table 3 and Fig. 12d, the trends of amplitude variation of both
eddies can be well predicted with the decreasing of AE1 and slow increase of AE2.
For AE1, the results of the prediction and observation are very close in the first two
weeks, with the centers of the two almost coinciding. The central position of the
prediction and observation began to deviate after the third week. For AE2, although
the amplitude and movement path are not predicted well at its initial stage, the
prediction is slowly approaching to the observation during third to fifth week, and
distance between the center of the prediction and the observation is reduced from 132

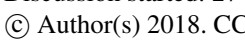



km at the beginning to 81 km at the end (see Fig. 12d the black line).

As mentioned above, the purpose of Exp5 is to investigate whether the decay of

AE1 and the continued development of AE2 can be predicted. From Fig. 10, Table 3
and Fig. 12e, we can find that the CSCASS cannot predict the movement path of AE1
well in its decay stage: the distance between the center of the prediction and that of
the observation is greater than 188 km, and the direction of movement is not
consistent (see red lines and dots in Fig. 10f). But the evolution and direction of
movement of AE2 can be well predicted at this stage. The amplitude of observation
and prediction of AE2 is almost constant (Fig. 12e), although the speed of movement
of AE2 given by prediction is slower than that of observation (see green lines and dots
in Fig. 10f).

The aim of Exp6 is to find whether the disappearance of AE1 and AE2 can be

both predicted. As described in Fig. 11, the disappearance of AE1 cannot be well
predicted since the low amplitude (less than 8 cm) of AE1 at this stage. Similarly, the
disappearance of AE2 is also less accurately predicted by the CSCASS (Fig. 12f). The
amplitude of AE2 from the observation decays continually at this stage, but the
amplitude of the predicted almost keeps constant. In addition, there is large deviation
of the direction of movement between prediction and observation for AE2 (see the red
solid line and dot in Fig. 11f).

**4. Conclusions and challenges for forecasting of mesoscale eddy**





In this paper, the reproduction and predictability of two representative
anticyclonic eddies, which have been observed in the NSCS, are investigated by a
series of assimilation and prediction experiments based on a Chinese Shelf/Coastal
Seas Assimilation System (CSCASS), along with the observations from surface
drifter trajectory and satellite remote sensing.
Quantitative and qualitative analyses of assimilation with the observations from
satellite remote sensing and drifter buoys shown that the generation and movement of
AE1 can be well reproduced by the CSCASS. In addition, the spatial pattern of AE1 is
also well reproduced by the CSCASS: the meridional and zonal radii of AE1 detected
by the assimilation (163 km and 93 km) are almost equivalent to that of observations
(148 km and 79 km). At the same time, the migration path of AE1 is well reproduced
by the CSCASS until its amplitude decays to less than 8 cm. In addition to AE1, the
evolution and propagation of AE2: moves firstly northwestward and then
southwestward, are well reproduced by the CSCASS, although large discrepancies
between model and observations are seem during its generation and decaying periods.
The comparisons of AE1 and AE2 from six predicted experiments with
observations show that the generation, evolution and movement path of these two
eddies with high amplitude (>8 cm) can be well predicted by the CSCASS, although
their generative mechanisms are quite different. The generated position and initial
migration of AE1 are well forecasted by the CSCASS, with amplitude matching well
with that of observation. The southwestward movement of AE1 along the continental
shelf with its amplitude decreasing and again increasing after its generation are also



predicted by the CSCASS. In addition, the first attenuation and then enhancement of
AE1 are well predicted by the CSCASS. On the whole, the development and
movement path of AE1 can be well predicted by CSCASS for the first four weeks
after its generation. After that, the errors between observation and prediction increase
significantly and by the fifth week, the distance betweem the prediction center and
that of observation become large and more than 100 km. The generation of AE2
cannot be predicted. This may be related to the lower amplitude (<8 cm) at this period.
The slow increase of AE2 from the second week after generation can be predicted,
with the prediction slowly approaching to the observation. During third to fifth week,
the amplitude of prediction of AE2 is almost equivalent to that of observation,
although the movement speed of the prediction is slower than that of observation.

In general, analyses of these two representative anticyclonic eddies in the NSCS

shown that generation, development and propagation of AE1 and AE2 can be well
reproduced and predicted by the CSCASS when their amplitude >8 cm. In contrast,
when their intensities are less than 8 cm, the generation and decay of these two
mesoscale eddies cannot be well reproduced and predicted by the system.

Since the mesoscale eddies are related to strong nonlinear processes and are not a

deterministic response to atmospheric forcing, the reproduction and predictability of
mesoscale eddies may depend mainly on the initial conditions of predicted system. In
addition, since the dynamical mechanisms of mesoscale eddies are quite different as
mentioned above, thus the ability of the ocean numerical model to represent the
physics and dynamics for mesoscale eddies is also crucial. Although data assimilation,



which combines observations with the numerical model, can provide good initial
conditions, it cannot make up the limitations of numerical model in numerical
algorithms and in its resolution. For a high-resolution operational oceanography, the
latter means that the numerical models need to be improved using more accurate
numerical algorithms especially in the weakly stratified regions or on the continental
shelf. So far most of the information about the ocean variability is mainly obtained
from satellites (SSH and SST), the information about the subsurface variability are
very rare. Although a substantial source of subsurface data is provided by the vertical
profiles (i.e.,expendable bathy thermographs, conductivity temperature depth, and
Argo floats), the datasets are still not sufficient to determine the state of the ocean. In
addition, in order to accurately assimilate the SSH anomalies from satellite altimeter
into the numerical model, it needs to know the oceanic mean SSH over the time
period of the altimeter observations (Xu et al., 2011; Rio et al., 2014). This is also a
big challenge because the earth's geoid is not presented with sufficient spatial
resolution when assimilating SSH in an eddy-resolving model. The future mission of
surface water and ocean topography (SWOT) launched in 2020 will help to resolve
and forecast the mesoscale features in eddy resolving ocean forecasting systems.
**Acknowledgements:**
This study is supported by the Marine Science and Technology Foundation of
South China Sea Branch, State Oceanic Administration (grant 1447), the National
Key Research and Development Program of China (2016YFC1401407), the Project of
Global Change and Air-Sea interaction under contract No. GASI-03-IPOVAI-04, the



National Natural Science Foundation of China (Grant No. 41776037 and 41276027),
and the China Scholarship Council (award to Xu Dazhi for 1 year's study abroad at
Nansen Environmental and Remote Sensing Center).

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




**Figures:**

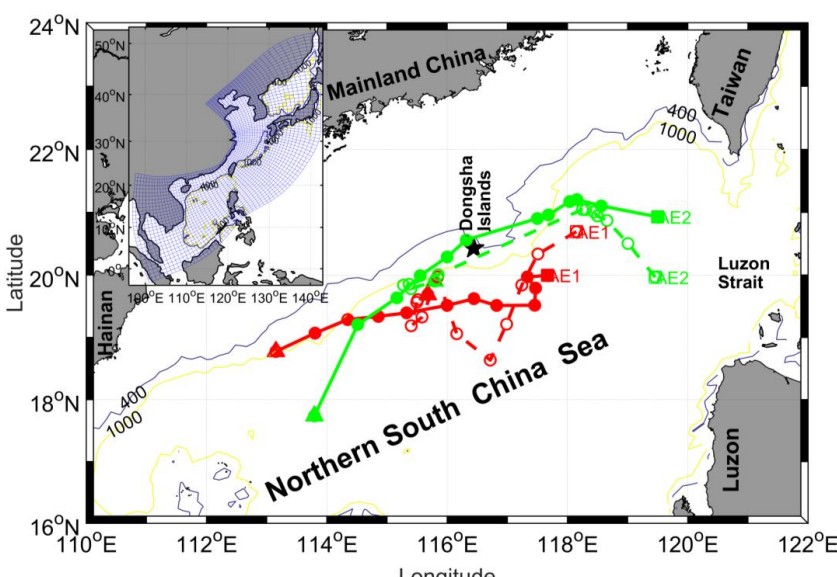


Fig. 1 Bathymetry of the northern South China Sea. The blue and yellow contour lines
are the isolines of 400 m and 1000 m. The solid black Pentagram indicated Dongsha
Islands. Red solid (hollow) circle dots and solid (dash) lines indicated weekly passing
position and migration path of observation (assimilation) AE1. Green solid (hollow)
circle dots and solid (dash) lines indicated weekly passing position and migration path
of observation (assimilation) AE2. The quadrangle and triangle denoted start and end
position, respectively. The model domain of CSCSS (the inset panel), the curvilinear
orthogonal model grid with 1/8-1/12° horizontal resolution (147×430) is denoted by
the blue grid (at intervals of 10 grid cells here).


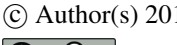



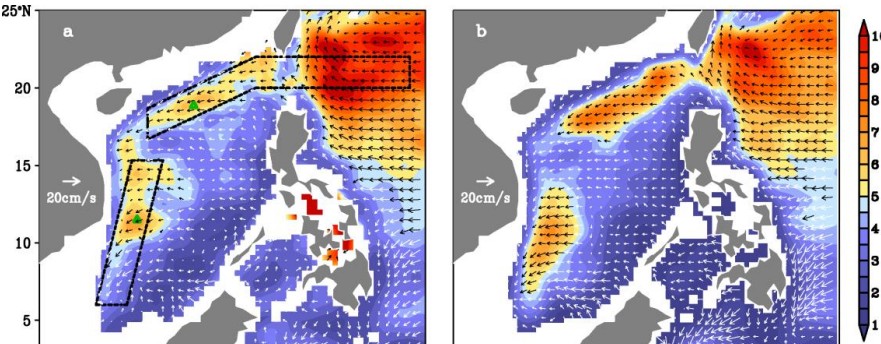

543 Fig. 2 Annual mean standard deviation of sea level mesoscale signals (color shading,

544 unit: cm) and propagation velocities of the signals (vectors) derived from (a) altimeter

545 observations; (b) OFES (OGCM for the Earth Simulator) simulations From Zhuang et

546 al. (2010).




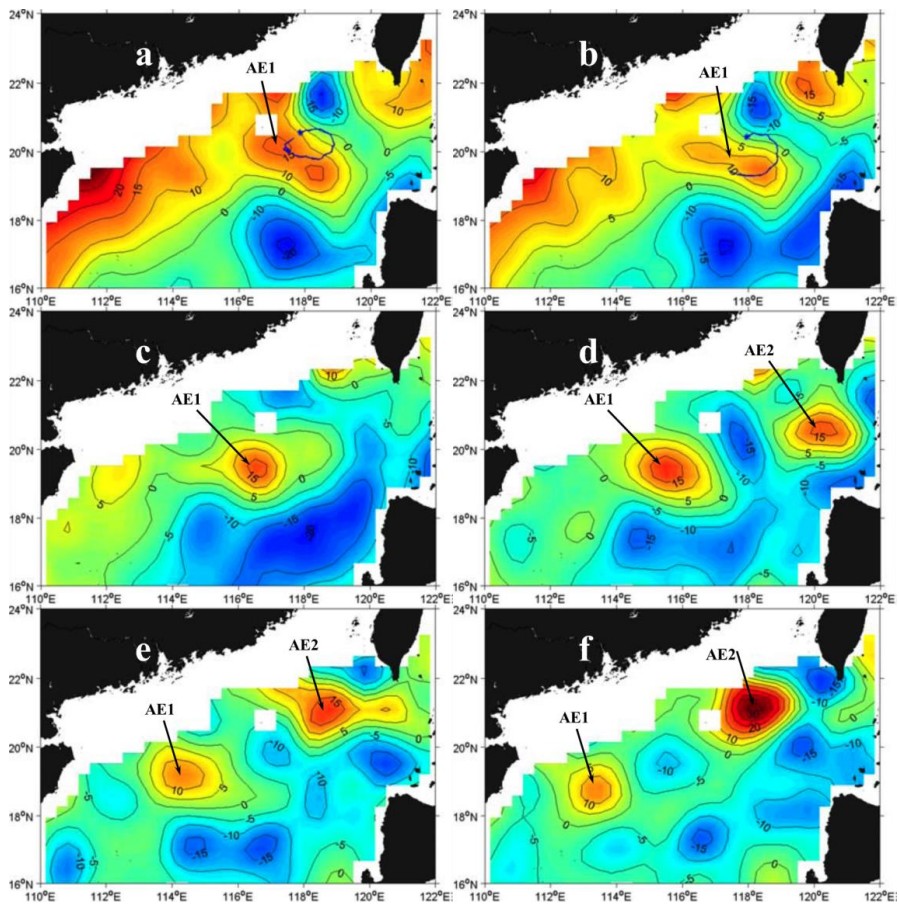


Fig. 3 Snapshots of SLA from satellite remote sensing datasets. Buoy 22918 trajectory
(blue lines, blue asterisk represents the initial position of buoy, as in Fig. 4) (a) from
December 4–15, 2003 superposed on SLA field on December 10, 2003; (b) from
December 16– 23, 2003 superposed on SLA field on December 17, 2003; SLA field
on (c) January 7, 2004; (d) January 21, 2004; (e) February 4, 2004; (f) February 18,
2004. From Wang et al. (2008).




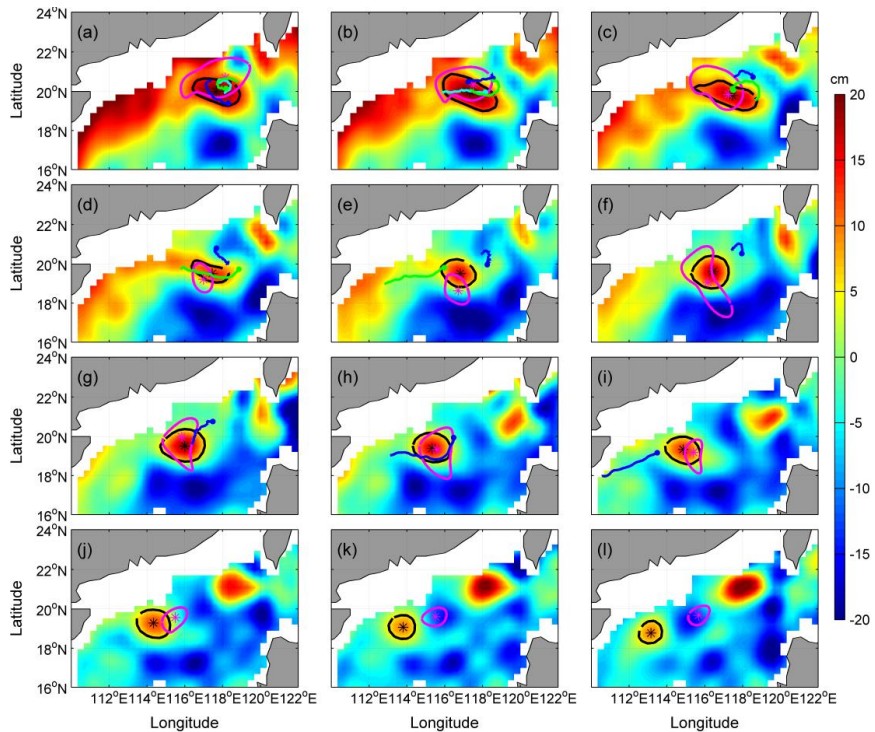


Fig. 4 Comparisons of AE1 derived from weekly SLA of assimilation results and
observation from satellite remote sensing during the period of December
2003~February 2004. Background color is SLA, "*" mark and closed lines indicated
the center position and the outermost closed isoline of AE1, respectively, the black is
from satellite observation SLA, the pink is from assimilation SLA. The cyan, green
and blue solid circle lines indicated the start positions and trajectories of drifter buoys
numbered 22517, 22918 and 22610, respectively. (a)-(l) is SLA on the 3rd of
December 2003 to 18th of February 2004, respectively. Unit: cm.



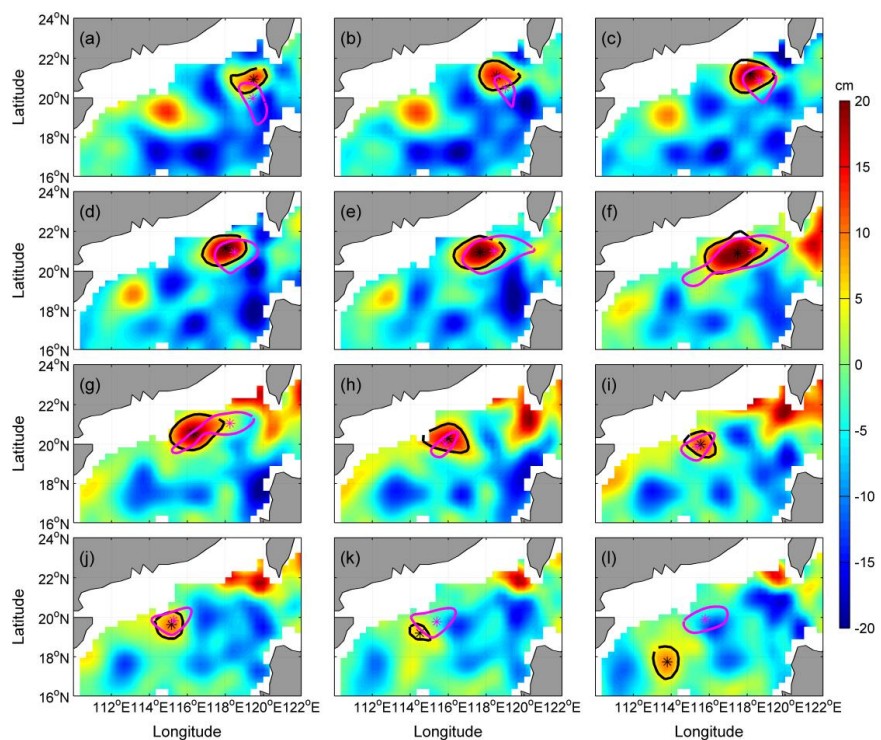


Fig. 5 The same as figure 4, But for AE2, the corresponding period is January 28th,
2003 to April 14th, 2003.



**Tables:**
Table 1 The settings of assimilation and six forecast experiments, including the start
and end date, the assimilation strategy of each experiment.

| Name | Start Date | End Date | Data Assimilated |
|------|-----------|----------|------------------|
| **As_exp** | 27/09/2003 | 02/05/2004 | SST+SLA every 3 days |
| **Exp1** | 29/11/2003 | 29/12/2003 | SST+SLA at first day |
| **Exp2** | 10/12/2003 (DAY0) | 09/01/2004 | SST+SLA at first day |
| **Exp3** | 31/12/2003 | 30/01/2004 | SST+SLA at first day |
| **Exp4** | 21/01/2004 | 20/02/2004 | SST+SLA at first day |
| **Exp5** | 08/02/2004 | 09/03/2004 | SST+SLA at first day |
| **Exp6** | 29/02/2004 | 30/03/2004 | SST+SLA at first day |






Table 2 The intensity and amplitude of AE1 and AE2 derived from observation SLA
and the assimilation SLA, and distance of eddy centers between the observation
SLA's and assimilation SLA's.

| Weekly | | | 1 | 2 | 3 | 4 | 5 | 6 | 7 | 8 | 9 | 10 | 11 | 12 |
|---|---|---|---|---|---|---|---|---|---|---|---|---|---|---|
| AE 1 | Distance (km) | | 94 | 4 | 2 | 6 | 9 | 7 | 5 | 3 | 6 | 13 | 19 | 29 |
| | Amplitude (cm) | Observed | 22 | 2 | 1 | 1 | 1 | 1 | 1 | 1 | 1 | 13 | 10 | 10 |
| | | Assimilate | 29 | 2 | 2 | 1 | 1 | 1 | 1 | 1 | 1 | 10 | 8 | 7 |
| | Intensity(cm) | Observed | 8 | 1 | 9 | 4 | 8 | 1 | 1 | 1 | 8 | 8 | 4 | 6 |
| | | Assimilate | 18 | 1 | 1 | 6 | 5 | 4 | 5 | 6 | 2 | 3 | 3 | 2 |
| AE 2 | Distance (km) | | 10 | 8 | 6 | 5 | 8 | 9 | 2 | 3 | 2 | 26 | 11 | 32 |
| | Amplitude (cm) | Observed | 14 | 1 | 2 | 2 | 2 | 2 | 2 | 1 | 1 | 11 | 6 | 10 |
| | | Assimilate | 8 | 1 | 1 | 1 | 2 | 1 | 1 | 1 | 1 | 15 | 12 | 11 |
| | Intensity (cm) | Observed | 7 | 1 | 1 | 1 | 1 | 1 | 1 | 1 | 7 | 6 | N/ | 6 |
| | | Assimilate | 3 | 2 | 5 | 6 | 1 | 8 | 4 | 8 | 9 | 4 | 5 | 6 |


Table 3 The intensity of AE1 and AE2 derived from observation SLA and the six
forecast SLA, and distance of eddy centers between the observation SLA's and
forecast SLA's.

| Weekly | | | | 1 | 2 | 3 | 4 | 5 |
|---|---|---|---|---|---|---|---|---|
| Exp1 | Intensity (cm) | | Observed | 8 | 10 | 9 | 8 | 8 |
| | | | Forecasted | 14 | 12 | 14 | 11 | 12 |
| Exp2 | Intensity (cm) | | Observed | 10 | 9 | 4 | 8 | 13 |
| | | | Forecasted | 12 | 11 | 6 | 8 | 10 |
| Exp3 | Intensity (cm) | | Observed | 13 | 13 | 11 | 8 | 8 |
| | | | Forecasted | 2 | 3 | 3 | 3 | N/A |
| Exp4 | AE1 | Intensity (cm) | Observed | 11 | 8 | 8 | 4 | 6 |
| | | | Forecasted | 4 | 2 | 2 | 2 | N/A |
| | AE2 | Intensity (cm) | Observed | N/A | N/A | 13 | 18 | 17 |
| | | | Forecasted | N/A | N/A | N/A | 6 | 9 |
| Exp5 | AE1 | Intensity (cm) | Observed | 4 | 6 | 2 | N/A | N/A |
| | | | Forecasted | 2 | 2 | 2 | 2 | 2 |
| | AE2 | Intensity (cm) | Observed | 18 | 17 | 17 | 17 | 14 |
| | | | Forecasted | 5 | 7 | 6 | 6 | 9 |
| Exp6 | AE2 | Intensity (cm) | Observed | 16 | 16 | 12 | 7 | 6 |
| | | | Forecasted | 7 | 9 | 6 | 4 | 6 |





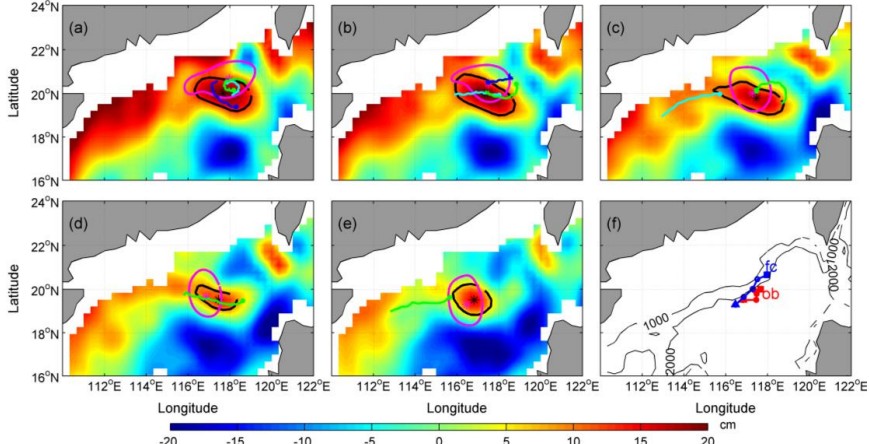

Fig. 6 Comparison of AE1 of Exp1 and observation, and trajectories of drifter buoys during the 29[th] of November 2003 to 29[th] of December 2004. The cyan, green and blue solid circle dots and lines indicated the start positions and trajectories of drifter buoys numbered 22917, 22918 and 22610 during the corresponding period, respectively. Where, the red (blue) dotted line in (f) is the moving path of AE1 derived from observation (forecast) SLA during the experiment period.

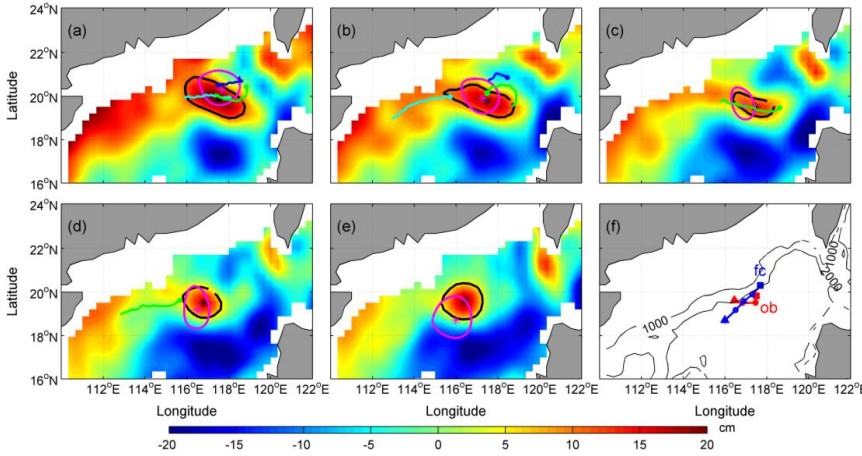

Fig. 7 Same as figure 6, but for Exp2, the experiment period is the 10[th] of December 2003 to the 9[th] of January 2004.




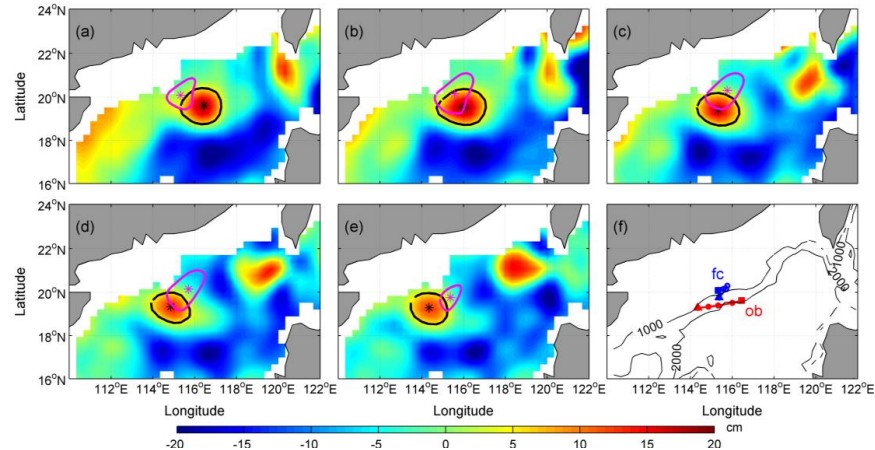


Fig. 8 Same as figure 7, but for Exp3, the experiment period is the 31st of December
2003 to the 30th of January 2004.


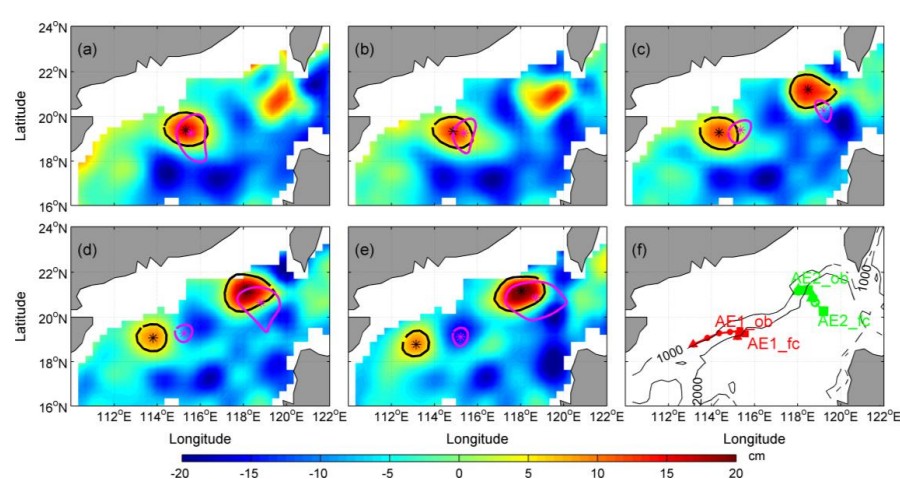


Fig. 9 Same as figure 8, but for Exp4,where, the red (green) dotted line in (f) is the
moving path of AE1 (AE2), the red solid lines and circle dots derived from
observation SLA, the green dash line and hollow circle dots derived from forecast
SLA during the 21st of January 2004 to the 20th of February 2004.





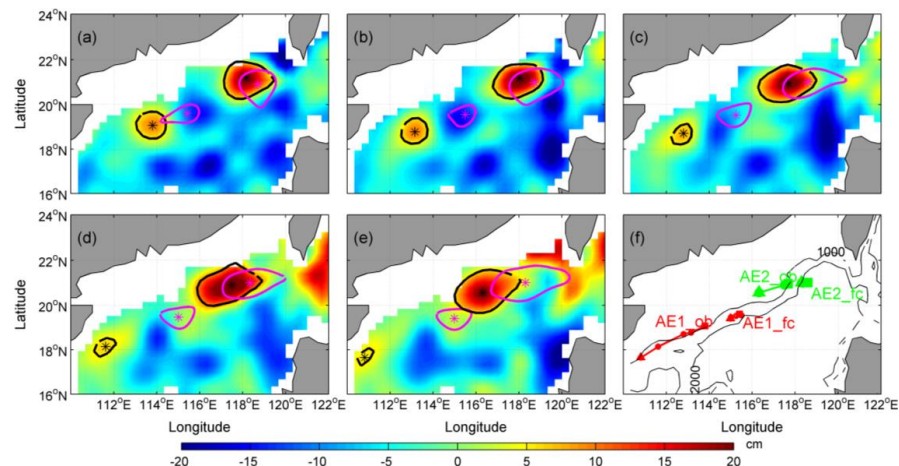


Fig. 10 Same as figure 9, but for Exp5, the experiment period is the 8th of February
2004 to the 10th of March 2004.



Fig. 11 Same as figure 9, but for Exp6 and AE2, the experiment period is the 29th of
February 2004 to the 30th of March 2004.




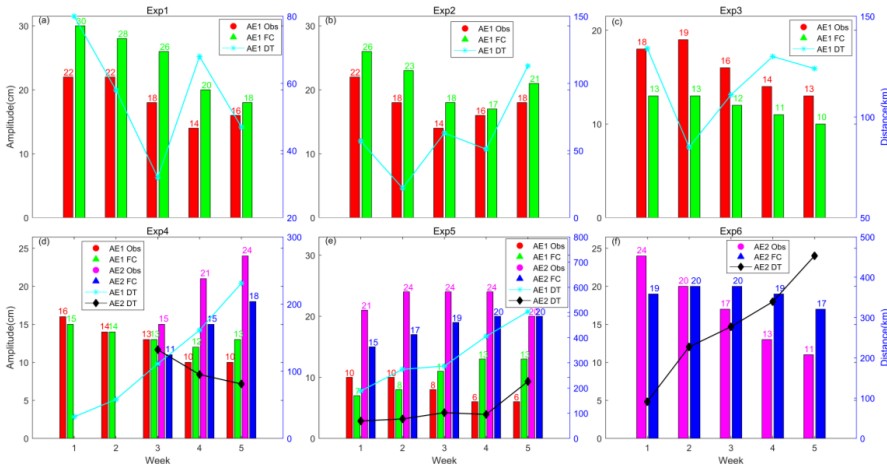

Fig. 12 The amplitude of AE1 and AE2 derived from observation SLA and the six forecast SLA, and distance of eddy centers between the observation SLA's and forecast SLA's. The red and green histograms indicated the amplitude of observation and prediction AE1. The pink and blue histograms expressed the amplitude of observation and prediction AE2. The cyan star line shows the distance of the center between observation and prediction AE1. The black diamond line shows the distance of the center between observation and prediction AE2.