# Peer review of "Could the two anticyclonic eddies during winter 2003/2004 be reproduced and"

_Ocean Science, 2018_

## Referee Comment (RC1) · Anonymous Referee #1 · 10 Sep 2018

The related studies about the mesoscale eddies in the SCS have amount of achievements, especially owing to the altimeter data widely applied, for understanding the dynamic and the interactions with the environmental current circulations on large scale. The article of "Could the mesoscale eddies be 1 reproduced and predicted in the northern south China sea: case studies" would like focus on two anticyclonic eddies in the northern SCS (NSCS). By helps of a HYCOM-EnOI assimilation system, they found the key of the predictable issues about the eddy generation, evolution and propagation paths can be done well only when the eddy amplitude is larger than 8 cm. Clearly, this topic is interesting for deep understanding the real factors to limit the eddy's forecast performance. The used methods, the related experiments, the main conclusions in

this study are creditable. But there are obvious some errors in text and figures/tables, this version needs to be more clear to state the findings and the concerned writings, although don't need to add more experiments.

The main comments and some found errors are listed as follow. 1) Under the current introduction, the reasons why to choose the two eddies in the north SCS are not clear enough. It means the necessity and the representative still need to be highlight. For example, complement the more details about these two eddies: the lifetime (Section 3.1); all the related references; methods and main points in Wang et al. (2008) and then relate to the aims in this study.

2) The eddy amplitude of 8 cm is a main finding in this study. For my opinion, it should have a relation with the SLA error in this system. Before the comparison of the eddy paths under different conditions, it is important to evaluate your simulated SLA (like in As_exp) to know how about the uncertainty. So one paragraph should be added.

3) It is important to clearly define how to objectively evaluate the eddy reproduction is well. In this study, the compared result is referred to the buoy trajectory and the detected by altimetry. Clearly, the related formula as possible can relate to these two elements. It will be helpful to simple and conclude in Table 2 and 3. For instance, P6 L230 "From Fig. 4 and Table 2, we can see that the generation and movement of AE1 can be well reproduced by the CSCASS . . ." add the related error statement and then objectively to know reproduced well or not.

4) P 1 L 51: ". . . high resolution satellite images or numerical model simulations (Yang et al., 2000), . . ." needs to add more reference about the recent key findings about mesoscale eddy both from satellite and modelling like as following:

Fu, L.-L., D.B. Chelton, P.-Y. Le Traon, and R. Morrow. 2010. Eddy dynamics from satellite altimetry. Oceanography 23(4):14–25, https://doi.org/10.5670/oceanog.2010.02.

Morrow, R. and Le Traon, P.-Y. Recent advances in observing mesoscale ocean dynamics with satellite altimetry. Adv. Spa. Res. 50, 1062–1076 (2012).

Frenger, I., Gruber, N., Knutti, R. & Münnich, M. Imprint of southern ocean eddies on winds, clouds and rainfall. Nat. Geosci 6, 608–612 (2013).

5) L 52: "… the operational forecasts of the mesoscale eddy still poses a big challenge because of its complicated dynamical mechanisms and high nonlinearity (Yuan and Wang, 1986; Li et al., 1998)." These references are not suitable because they are not related with ocean operational forecast and were published more than 20 years out of representing the recent knowledge.

Some references are recommended as follow: De Vos, M., Backeberg, B. and Counillon, F.: Using an eddy-tracking algorithm to understand the impact of assimilating altimetry data on the eddy characteristics of the Agulhas system. Ocean Dyn., https://doi.org/10.1007/s10236-018-1174-4, 2018.

Robert H. Woodham, Oscar Alves, Gary B. Brassington, Robin Robertson & Andrew Kiss (2015) Evaluation of ocean forecast performance for Royal Australian Navy exercise areas in the Tasman Sea, Journal of Operational Oceanography, 8:2, 147-161, DOI: 10.1080/1755876X.2015.1087187

Treguier Anne-Marie, Chassignet Eric P., Le Boyer Arnaud, Pinardi Nadia (2017). Modeling and forecasting the "weather of the ocean" at the mesoscale. J. Marine Research, 75(3), 301-329. http://doi.org/10.1357/002224017821836842.

6) P3 L87: "… thus is essential for the prediction of mesoscale eddies (e.g., Xiao et al. 2007; Xie et al., 2011; Xu et al., 2012; Xie et al., 2018)". The concerned assimilation works done in the NSCS needs be commented, and then to be pointed the disadvantages to relate the aims in this study.

Xiao, X., Wang, D., Yan, C., and Xu, J.: The assimilation experiment in the southwestern South China Sea in summer 2000, Chinese Sci. Bull., 51, 31–37, 2007.

Xie, J., Bertino, L., Cardellach, E., Semmling, M., and Wickert, J.: An OSSE evaluation of the GNSS-R altimetry data for the GEROS-ISS mission as a complement to the existing observational networks, Remote Sens. Environ., 209, 152-165, doi:10.1016/j.rse.2018.02.053, 2018.

Xie, J., Counillon, F., Zhu, J., and Bertino, L.: An eddy resolving tidal-driven model of the South China Sea assimilating along-track SLA data using the EnOI, Ocean Sci., 7, 609–627, doi:10.5194/os-7-609-2011, 2011

7) P5 L131: Are there some cases using this detection scheme in the SCS? Yes, give the reference, otherwise provide a simple snapshot to show its ability.

8) Table 1 lists the designed experiment time. For instance (my personal point), the experiments designed by the eddy strength should be highlighted using one figure to replace the table. On this figure, the eddy strengths of AE1 and AE2 are curved as a function of the date, and the experimental date at beginning also are marked on by vertical lines.

9) Table 2: The dates of the first weeks need to be stated. What the differences between "Amplitude" and "Intensity"? As the statement of P4 L127 "the intensity of the mesoscale eddy must be greater than 2 cm;", how the observed amplitudes of AE1/AE2 less than 2 cm? Are they the error or others? And to compare the amplitudes in the first and the second weeks, can comment the big gap?

10) Use the same color in the panel f of Figure 9-11 as the other panels of Fig. 6-8: the blue (red) is forecast (observation), and using full or empty mark to distinguish AE1 and AE2.

11) There are interested finding in Figure 12: at the first stage of AE1 and AE2 the distance error looks decreasing; at end stages the distance error increasing with time. Can you explain the former?

12) In Figure12, add another referenced eddy distance line from As_exp. It will be

interesting to compare these two lines to show the predictability if without data assimilation.

13) Recommend to replace the title by "Could the two anticyclonic eddies during winter 2003/2004 be reproduced and predicted in the northern south China sea?"

Technic comments:

Figure 3 is too ambiguous.

P1, L62: "... (Fig. 1). Forced ..." the intensity of the mesoscale eddy must be greater than 2 cm;

P5: The paragraph introduces the ocean model should be shorten like deleting the lines of 140-150.

P7 L 170: "... as a surface forcing from Legates and Willmott (1990)." Legates, D.R., Willmott, C.J., 1990: Mean seasonal and spatial variability in gauge-corrected, global precipitation. Int. J. Climatology, 10, 111-127.

P7, L172: missing the reference of "Han (1984)".

P7, L 183: EnKF as the first place should give the detailed name.

P9, Section 3.1: The AE2 lifetime was not clearly stated so the first (last) identified date needs be mentioned.

Table 3: "... distance of eddy centers between the observation SLA's ..." are missing on the content. So double cheek the consistence in caption.

Figure 12: The cyan line is hard to see so change it to be black. The histogram should use the rectangle to present well other than circle and triangle. L631:"The red and green histograms indicated the AE1 amplitudes from observation and prediction respectively."

The wrong order of the references is clear like: P18 L 414 Bleck et al. (2002); P18

L421 Counillon and Bertino (2009); P18 L433 Hamilton et al. (1999); P19 L444 Kara et al. (2002); P20 L475 Rio et al. (2014); P20 L487 Woodruff et al. (1987)

---

## Referee Comment (RC2) · Anonymous Referee #2 · 16 Sep 2018

The motion and transport of mesoscale eddies have been intensively studied with the altimeter data. In contrast, the simulations are relatively very few, although they are more useful for prediction and applications. As the simulations in this paper are quite well, I have a few minor comments on results. My major concerns are how to improve the writing skill of the paper to make it more comprehensible and valuable for readers. In final, the result is interesting and valuable, but some minor revisions are required before publication.

1. In this paper, both amplitude and intensity are used. In general, eddy amplitude was common used in previous studies (e.g., Chelton et al., 2011). I suggest authors use

amplitude other than intensity in the paper.

2. The motivation of study may be stressed in a more comprehensive way for board readers, if the authors include the previous knowledge on evolution and propagation of oceanic eddies from altimeter data. The motion of mesoscale eddies would be a straight line, if eddies freely propagate in open ocean. However, most of eddies may have interaction with topography (costal and islands), strong currents (e.g., western boundary current), eddies during their lifetime. The motion of eddy will be modified and even split when approaching an island (Yang et al., 2017). It is also recognized that western boundary is graveyard of eddies (Zhai et al., 2010). The dynamical processes such as splitting and/or merging of eddies can also make termination and/or genesis of eddies in open ocean (Li et al., 2016). Thus the dynamical processes make that the prediction of eddy motion is a challenge for ocean simulation.

3. The result is useful that generation, evolution and propagation paths of AE1 and AE2 can be well reproduced and forecasted when their amplitude >8 cm. I have two comments on this point. Firstly, authors should clearly point out what "their amplitude" means, observed one or simulated one. Secondly, the values in tables should be clearly consist with this result.

4. Moreover, amplitude is good criterion, a dimensionless one might be better, which makes the result more valuable. This could be achieved if the authors may go one step further. As we know that mesoscale eddies are nonlinear compare with linear Rossby waves (Chelton et al., 2011), they are quite different, e.g., for propagation speed. It is hypothesized that the advective nonlinearity parameter might be presumably important, and authors may use it as an additional criterion. The advective nonlinearity parameter is defined as the nondimensional ratio U/c, where U is the maximum rotational speed and c is the translation speed of the eddy. A value of U/c> 1 implies theoretically that there is trapped fluid within the eddy interior that is advected with the eddy as the eddy translates, which is a fundamental distinction between linear waves and nonlinear eddies. The authors can check their results: what U/c exactly is in their simulations.

Others

Table 3, intensity –> amplitude

The labels AE1 and AE2 in Figure 1 are coved by symbols. Please shift them away from the symbols, and similar change for Figures 6, 9-11.

Line 576-579, the order of parameters in table caption are different from that in table. Please modify reorder the parameters.

References:

Chelton, D.B; Schlax, M.G., and Samelson, R.M., 2011, Global observations of nonlinear mesoscale eddies. Progress in Oceanography. 91, 167–216.

Li, Q.-Y., Sun, L., and Lin, S.-F.: GEM: a dynamic tracking model for mesoscale eddies in the ocean, Ocean Sci., 12, 1249-1267, https://doi.org/10.5194/os-12-1249-2016, 2016.

Yang, S., Xing, J., Chen, D., and Chen, S.: A modelling study of eddy-splitting by an island/seamount, Ocean Sci., 13, 837-849, https://doi.org/10.5194/os-13-837-2017, 2017.

Zhai X, Johnson H L, Marshall D P. Significant sink of ocean-eddy energy near western boundaries. Nature Geoscience, 2010, 3(9):608-612.

---

## Author Response (AR1)

Editor, Ocean Science
Email: editorial@copernicus.org

December 4, 2018

Dear Editor:

Thank you for your letter dated September 10 and 16 regarding our manuscript entitled "Could the mesoscale eddies be reproduced and predicted in the northern south China sea: case studies" (No: os-2018-74), which was submitted to Ocean Science for consideration of publication. We have read reviewers' comments carefully. Following your suggestions, we have gone through the manuscript and made the following changes in the manuscript.

The line-by-line replies for reviewers' comments are shown in the attachments. The original comments are quoted in Times New Roman (Bold) and our responses are in Times New Roman.

Once again, thank you and reviewers for the time and energy spending on reading our manuscript and providing constructive comments and suggestions, which are very valuable in improving the quality of our manuscript.

Enclosed, please find our updated manuscript and reply to the reviewers' comments.

We hope this version of our manuscript can meet the standard of publication in this journal, and we look forward to receiving further instruction related to this submission.

Sincerely yours

Dazhi Xu

**Responses to Reviewer 1**

**General comment:**

**The related studies about the mesoscale eddies in the SCS have amount of achievements, especially owing to the altimeter data widely applied, for understanding the dynamic and the interactions with the environmental current circulations on large scale. The article of "Could the mesoscale eddies be reproduced and predicted in the northern south China sea: case studies" would like focus on two anticyclonic eddies in the northern SCS (NSCS). By helps of a HYCOM-EnOI assimilation system, they found the key of the predictable issues about the eddy generation, evolution and propagation paths can be done well only when the eddy amplitude is larger than 8 cm. Clearly, this topic is interesting for deep understanding the real factors to limit the eddy's forecast performance. The used methods, the related experiments, the main conclusions in this study are creditable. But there are obvious some errors in text and figures/tables, this version needs to be more clear to state the findings and the concerned writings, although don't need to add more experiments.**

**The main comments and some found errors are listed as follow:**

**Ans:** We greatly appreciate reviewer for the time and energy spending on reading our manuscript and providing constructive comments and suggestions. We totally agree with the reviewer and we made every effort to clarify our results and improve the manuscript. The revised version reflects these changes. The detailed comments have been replied one by one below. Once again, thank you very much for your significant comments and suggestions, which are valuable in improving the quality of our manuscript.

**1) Under the current introduction, the reasons why to choose the two eddies in the north SCS are not clear enough. It means the necessity and the representative still need to be highlight. For example, complement the more details about these two eddies: the lifetime (Section 3.1); all the related references; methods and main points in Wang et al. (2008) and then relate to the aims in this study.**

**Ans:** Thank you for these comments. The introduction has been revised, the lifetime, evolution and propagation of these two eddies has been described, farther, the related reference is also been added in the revised versions. (P4 line 61-76).

**2) The eddy amplitude of 8 cm is a main finding in this study. For my opinion, it should have a relation with the SLA error in this system. Before the comparison of the eddy paths under different conditions, it is important to evaluate your simulated SLA (like in As_exp) to know how about the uncertainty. So one paragraph should be added.**

**Ans:** Thank you. The paragraph which describe the evaluation of the uncertainty of the CSCASS has been added in the revised versions. (P12, line 227-233).

**3) It is important to clearly define how to objectively evaluate the eddy reproduction is well.**

In this study, the compared result is referred to the buoy trajectory and the detected by altimetry. Clearly, the related formula as possible can relate to these two elements. It will be helpful to simple and conclude in Table 2 and 3. For instance, P6 L230 "From Fig. 4 and Table 2, we can see that the generation and movement of AE1 can be well reproduced by the CSCASS . . ." add the related error statement and then objectively to know reproduced well or not.

**Ans:** Thank you for your constructive advice. In the revised version, we used a dimensionless index called advective nonlinearity parameter (ANP, Chelton et al., 2011, Li et al., 2014; 2015; 2016; Wang et al., 2015), which expressed as the maximum rotational speed U divided by the translation speed c of the eddy, that is U/c (P12, line 234-237). As Fig.5 shows, if the U/c > 2 the CSCASS can well reproduce AE2 (P28).

**4) P 1 L 51: ". . . high resolution satellite images or numerical model simulations (Yang et al., 2000), . . ." needs to add more reference about the recent key findings about mesoscale eddy both from satellite and modelling like as following:**

**Fu, L.-L., D.B. Chelton, P.-Y. Le Traon, and R. Morrow. 2010. Eddy dynamics from satellite altimetry. Oceanography 23(4):14–25, https://doi.org/10.5670/oceanog.2010.02.**

**Morrow, R. and Le Traon, P.-Y. Recent advances in observing mesoscale ocean dynamics with satellite altimetry. Adv. Spa. Res. 50, 1062–1076 (2012).**

**Frenger, I., Gruber, N., Knutti, R. & Münnich, M. Imprint of southern ocean eddies on winds, clouds and rainfall. Nat. Geosci 6, 608–612 (2013).**

**Ans:** Thank you. The references have been added in the revised versions. (P3, line 40-41).

**5) L 52: ". . . the operational forecasts of the mesoscale eddy still poses a big challenge because of its complicated dynamical mechanisms and high nonlinearity (Yuan and Wang, 1986; Li et al., 1998)." These references are not suitable because they are not related with ocean operational forecast and were published more than 20 years out of representing the recent knowledge.**

**Some references are recommended as follow: De Vos, M., Backeberg, B. and Counillon, F.: Using an eddy-tracking algorithm to understand the impact of assimilating altimetry data on the eddy characteristics of the Agulhas system. Ocean Dyn., https://doi.org/10.1007/s10236-018-1174-4, 2018.**

**Robert H. Woodham, Oscar Alves, Gary B. Brassington, Robin Robertson & Andrew Kiss (2015) Evaluation of ocean forecast performance for Royal Australian Navy exercise areas in the Tasman Sea, Journal of Operational Oceanography, 8:2, 147-161, DOI: 10.1080/1755876X.2015.1087187**

**Treguier Anne-Marie, Chassignet Eric P., Le Boyer Arnaud, Pinardi Nadia (2017). Modeling and forecasting the "weather of the ocean" at the mesoscale. J. Marine Research, 75(3), 301-329. http://doi.org/10.1357/002224017821836842.**

**Ans:** Thank you. The references have been added in the revised versions. (P3, line 43)

**6) P3 L87: ". . . thus is essential for the prediction of mesoscale eddies (e.g., Xiao et al. 2007; Xie et al., 2011; Xu et al., 2012; Xie et al., 2018)". The concerned assimilation works done in the NSCS needs be commented, and then to be pointed the disadvantages to relate the aims in this study.**

**Xiao, X., Wang, D., Yan, C., and Xu, J.: The assimilation experiment in the southwestern South China Sea in summer 2000, Chinese Sci. Bull., 51, 31–37, 2007.**

**Xie, J., Bertino, L., Cardellach, E., Semmling, M., and Wickert, J.: An OSSE evaluation of the GNSS-R altimetery data for the GEROS-ISS mission as a complement to the existing observational networks, Remote Sens. Environ., 209, 152-165, doi:10.1016/j.rse.2018.02.053, 2018.**

**Xie, J., Counillon, F., Zhu, J., and Bertino, L.: An eddy resolving tidal-driven model of the South China Sea assimilating along-track SLA data using the EnOI, Ocean Sci., 7, 609–627, doi:10.5194/os-7-609-2011, 2011.**

**Ans:** Thank you. The references have been added in the revised versions. (P5, line 91)

**7) P5 L131: Are there some cases using this detection scheme in the SCS? Yes, give the reference, otherwise provide a simple snapshot to show its ability.**

**Ans:** Yes, Cheng et al., (2005) used this detection scheme to study the seasonal and interannual variabilities of mesoscale eddies in South China Sea. (P7, line 125)

**8) Table 1 lists the designed experiment time. For instance (my personal point), the experiments designed by the eddy strength should be highlighted using one figure to replace the table. On this figure, the eddy strengths of AE1 and AE2 are curved as a function of the date, and the experimental date at beginning also are marked on by vertical lines.**

**Ans:** Thank you. According to your advice, we use Fig.4 to replace table 1. (P28, Fig.4)

**9) Table 2: The dates of the first weeks need to be stated. What the differences between "Amplitude" and "Intensity"? As the statement of P4 L127 "the intensity of the mesoscale eddy must be greater than 2 cm;", how the observed amplitudes of AE1/AE2 less than 2 cm? Are they the error or others? And to compare the amplitudes in the first and the second weeks, can comment the big gap?**

**Ans:** Thank you. The dates of the first weeks have been added to table 2 (P31, table1); In the original version, the intensity is the difference between the extremum and the outermost closure of SLA; the amplitude is the difference between the extremum and 0 of SLA. In the revised version, the amplitude is the difference between the extremum and the outermost closure of SLA, and do not use the intensity. The observed amplitudes of AE1/AE2 less than 2 cm are errors and corrected in the revised version (P31, table1).

**10) Use the same color in the panel f of Figure 9-11 as the other panels of Fig. 6-8: the blue (red) is forecast (observation), and using full or empty mark to distinguish AE1 and AE2.**

**Ans:** Thank you. The related figures have been revised in the revised versions (P34, Fig.11; P35, Fig.12-13).

**11) There are interested finding in Figure 12: at the first stage of AE1 and AE2 the distance error looks decreasing; at end stages the distance error increasing with time. Can you explain the former?**

**Ans:** Thank you! As our results show, at the first stage of AE1 and AE2, they are in strong intensity stage or become more and more strong. The CSCASS, after assimilated SLA and SST, can well reproduce these eddies. But at the end stages, the signals of eddy become weak, so the CSCASS can not catch even assimilated the SLA and SST.

**12) In Figure12, add another referenced eddy distance line from As_exp. It will be interesting to compare these two lines to show the predictability if without data assimilation.**

**Ans:** Thank you. The eddy distance line from As_exp has been added to Fig.12 in the revised versions. (P36, Fig.14)

**13) Recommend to replace the title by "Could the two anticyclonic eddies during winter 2003/2004 be reproduced and predicted in the northern south China sea?"**

**Ans:** Thank you. The title has been revised in the revised versions. (P1, line 1-2)

**Technic comments:**

**Figure 3 is too ambiguous.**

**Ans:** Thank you. This figure has been changed in the revised versions. (P27, Fig.3)

**P1, L62: "... (Fig. 1). Forced . . .";**

**Ans:** Thank you. The word 'Forcing' has been changed to 'Forced' in the revised versions. (P3, line 50)

**the intensity of the mesoscale eddy must be greater than 2 cm;**

**Ans:** Thank you. You are right, the intensity of the mesoscale eddy is greater than 2 cm. There is a technical error in the original version, which leads to the intensity of the mesoscale eddy less than 2 cm. This error has been corrected in the revised versions. (P31, table 1)

**P5: The paragraph introduces the ocean model should be shorten like deleting the lines of 140-150.**

**Ans:** Thank you. The sentences have been deleted in the revised versions. (P8, line 141-150)

**P7 L170: ". . . as a surface forcing from Legates and Willmott (1990)." Legates, D.R., Willmott, C.J., 1990: Mean seasonal and spatial variability in gauge-corrected, global precipitation. Int. J. Climatology, 10, 111-127.**

**Ans:** Thank you. The reference has been corrected in the revised versions. (P22, line 451-452)

**P7, L172: missing the reference of "Han (1984)".**

**Ans:** Thank you. The reference has been added in the revised versions. (P22, line 438-439)

**P7, L 183: EnKF as the first place should give the detailed name.**

**Ans:** Thank you. The detailed name of EnKF has been added in the revised versions. (P10, line 182)

**P9, Section 3.1: The AE2 lifetime was not clearly stated so the first (last) identified date needs be mentioned.**

**Ans:** Thank you. The first (last) identified date of AE2 has been added in the revised versions. (P4, line 69, line 73)

**Table 3: ". . . distance of eddy centers between the observation SLA's . . ." are missing on the content. So double cheek the consistence in caption.**

**Ans:** Thank you. The distance of eddy centers for forecast experiments have been added in the revised versions. (P32, table 2)

**Figure 12: The cyan line is hard to see so change it to be black. The histogram should use the rectangle to present well other than circle and triangle. L631:"The red and green histograms indicated the AE1 amplitudes from observation and prediction respectively."**

**Ans:** Thank you. The sentence has been corrected; The circle and triangle have been replaced by the rectangle in the new figure 12 (P36, Fig. 14). Due to the black line has been used to denote AE2, we still use the cyan line denote to AE1 in the revised versions.

**The wrong order of the references is clear like: P18 L 414 Bleck et al. (2002); P18 L421 Counillon and Bertino (2009); P18 L433 Hamilton et al. (1999); P19 L444 Kara et al. (2002); P20 L475 Rio et al. (2014); P20 L487 Woodruff et al. (1987)**

**Ans:** Thank you. The order of the references has been corrected in the revised versions. (P21-25, line 410-533)

**Responses to Reviewer 2**

**General comments:**

**The motion and transport of mesoscale eddies have been intensively studied with the altimeter data. In contrast, the simulations are relatively very few, although they are more useful for prediction and applications. As the simulations in this paper are quite well, I have a few minor comments on results. My major concerns are how to improve the writing skill of the paper to make it more comprehensible and valuable for readers. In final, the result is interesting and valuable, but some minor revisions are required before publication.**

**Ans:** Thank you very much for your supports and valuable comments. We totally agree with the reviewer and we made every effort to clarify our results and improve the manuscript. The revised version reflects these changes. The detailed comments have been replied one by one below. Once again, thank you very much for your significant comments and suggestions, which are valuable in improving the quality of our manuscript.

**1. In this paper, both amplitude and intensity are used. In general, eddy amplitude was common used in previous studies (e.g., Chelton et al., 2011). I suggest authors use amplitude other than intensity in the paper**

**Ans:** We totally agree with your comments. The intensity has been removed or changed to amplitude in the revised versions. (P7, line 129; P13, line 258; P14, line 288; P15, line 292; P16, line 317; P18, line 373)

**2. The motivation of study may be stressed in a more comprehensive way for board readers, if the authors include the previous knowledge on evolution and propagation of oceanic eddies from altimeter data. The motion of mesoscale eddies would be a straight line, if eddies freely propagate in open ocean. However, most of eddies may have interaction with topography (costal and islands), strong currents (e.g., western boundary current), eddies during their lifetime. The motion of eddy will be modified and even split when approaching an island (Yang et al., 2017). It is also recognized that western boundary is graveyard of eddies (Zhai et al., 2010). The dynamical processes such as splitting and/or merging of eddies can also make termination and/or genesis of eddies in open ocean (Li et al., 2016). Thus the dynamical processes make that the prediction of eddy motion is a challenge for ocean simulation.**

**Ans:** We greatly appreciate your support and constructive comments on our work. We agree with your comments. Thank you for the supportive and constructive comments on our manuscript.

**3. The result is useful that generation, evolution and propagation paths of AE1 and AE2 can be well reproduced and forecasted when their amplitude >8 cm. I have two comments on this point. Firstly, authors should clearly point out what "their amplitude" means, observed one or simulated one. Secondly, the values in tables should be clearly consist with this result.**

**Ans:** Thank you. The means of "their amplitude" has been clearly point out, it is the observed amplitude (P2, line 21); There is a technical error in the original version, which leads to the values in tables not consist with the result, the values in the table (P31, table 1) have been corrected in the revised versions.

**4. Moreover, amplitude is good criterion, a dimensionless one might be better, which makes the result more valuable. This could be achieved if the authors may go one step further. As we know that mesoscale eddies are nonlinear compare with linear Rossby waves (Chelton et al., 2011), they are quite different, e.g., for propagation speed. It is hypothesized that the advective nonlinearity parameter might be presumably important, and authors may use it as an additional criterion. The advective nonlinearity parameter is defined as the nondimensional ratio U/c, where U is the maximum rotational speed and c is the translation speed of the eddy. A value of U/c> 1 implies theoretically that there is trapped fluid within the eddy interior that is advected with the eddy as the eddy translates, which is a fundamental distinction between linear waves and nonlinear eddies. The authors can check their results: what U/c exactly is in their simulations.**

**Ans:** Thank you for your comments. The advective nonlinearity parameter U/c has been calculated, the results have been shown in Fig.5 (P28) in the revised versions.

**Others**

**Table 3, intensity –> amplitude**

**Ans:** Thank you. The word "intensity" has been changed to "amplitude" in the revised versions. (P32, table 2)

**The labels AE1 and AE2 in Figure 1 are coved by symbols. Please shift them away from the symbols, and similar change for Figures 6, 9-11.**

**Ans:** Thank you. The related figures have been corrected in the revised versions. (P26, Fig.1; P33-35, Fig.8-13)

**Line 576-579, the order of parameters in table caption are different from that in table. Please modify reorder the parameters.**

**Ans:** Thank you. The table has been corrected in the revised versions. (P32, table 2)

[revised manuscript text omitted]
 is as follows: 1) there must be a closure contour on the merged SLA; 2) there must have 
[revised manuscript text omitted]

---

## Author Response (AR2)

Editor, Ocean Science

Email: editorial@copernicus.org

December 22, 2018

Dear Editor:

Thank you for your letter dated December 12 regarding our manuscript entitled "**Could the mesoscale eddies be reproduced and predicted in the northern south China sea: case studies**" (No: os-2018-74), which was submitted to **Ocean Science** for consideration of publication. We have read reviewer's comments carefully. Following your suggestions, we have gone through the manuscript and made the following changes in the manuscript.

The line-by-line replies for reviewer's comments are shown in the attachments. The original comments are quoted in Times New Roman (Bold) and our responses are in Times New Roman.

Once again, thank you and reviewer for the time and energy spending on reading our manuscript and providing constructive comments and suggestions, which are very valuable in improving the quality of our manuscript.

Enclosed, please find our updated manuscript and reply to the reviewer's comments.

We hope this version of our manuscript can meet the standard of publication in this journal, and we look forward to receiving further instruction related to this submission.

Sincerely yours

Dazhi Xu

**Topic Editor Decision: Publish subject to minor revisions (review by editor) (12 Dec 2018) by John M. Huthnance**

**Comments to the Author:**

**Dear Authors**

**Thank-you for your revisions. I think some of the review comments are "open to interpretation" and so am still asking for some "Minor Revision". This is for clarity and to include more of what I think the reviewers were asking for. My comments follow here.**

**Ans:** We are very grateful to the editor for the recognition of our work, and we are very grateful for your efforts and time to improve the quality of this article.

**Comments**

**Page 3**

**Lines 31-32. Better ". . atmosphere, ocean mesoscale eddies are often described . ."**

**Ans:** Thank you. The sentences have been revised in the revised version. (P3, line 31-32)

**Lines 39-40. You have added the references but I think Referee 1 wanted a few words about what is learned from these references, e.g. scales resolved (Fu et al), eddy energy and tracking (Morrow and Le Traon), effects on atmosphere (Frenger et al.).**

**Ans:** Thank you for your constructive advice. The sentences have been rewrote to adapt to the added references in the revised version. (P3, line 45-47)

**Line 42 (or nearby). I am sure Referee 2 comment 2 wants more text about motivation in your manuscript, not just the references. Their comment is a suggestion for what to include.**

**Ans:** Thank you. These contents have been added in the revised version. (P3, line 37-45)

**Page 3 line 51 to Page 4 line 1. Better ". . exhibits significant mesoscale eddy activity (Fig. 2). Many studies have tried to investigate mesoscale eddies in the NSCS . ."**

**Ans:** Thank you. The sentences have been revised in the revised version. (P4, line 61-62)

**Page 4**

**Line 57. ". . recorded evidence . ." (omit "the")**

**Ans:** Thank you. The word "the" has been deleted in the revised version. (P4, line 67)

**Line 72. ". . 2004. Meanwhile, . ."**

**Ans:** Thank you. The sentences have been revised in the revised version. (P5, line 82)

**Line 73. ". . disappeared southeast of Hainan . ."**

**Ans:** Thank you. The sentences have been revised in the revised version. (P5, line 83)

**Page 5**

**Lines 78-79. ". . Despite the activities . . NSCS having received . ."**

**Ans:** Thank you. The sentences have been revised in the revised version. (P5, line 87-88)

**Line 82. ". . (Oey et al., 2005); they are . ."**

**Ans:** Thank you. The sentences have been revised in the revised version. (P5, line 91)

**Line 87. You have added the references suggested by Referee 1. However, I think you also need to comment on what they show (achieve), what is still needed and so how your study helps.**

**Ans:** Thank you. The contents have been added in the revised version. (P6, line 96-103)

**Line 88. ". . two typical anticyclonic eddies . ." Are these AE1 and AE2 (not clear as you have re-arranged the text)? Referee 1 (main comment 1) asked why these two particular eddies were chosen; you should say (they ought to be representative, survive long enough to be useful . . . etc.).**

**Ans:** Thank you. The contents have been added in the revised version. (P6, line 104-105)

**Page 6**

**Line 112. ". . System, are also used. . ." It would be better to divide this sentence which is too long.**

**Ans:** Thank you. The sentence has been revised in the revised version. (P7, line 129)

**Line 116. ". . used. Three drifters were designed"**

**Ans:** Thank you. The sentence has been revised in the revised version. (P7, line 133-134)

**Page 7**

**Lines 122-126. ". . study as follows: 1) there must be a closed contour on the merged SLA; 2) there must be one maximum or minimum inside the area of the closed contour for anticyclonic or cyclonic eddy; 3) the difference between the extremum and the outermost closed SLA contour, that is, the amplitude of the mesoscale eddy, must be greater than 2 cm; . ."**

**Ans:** Thank you. The sentences have been revised in the revised version. (P8, line 140-144)

**Line 136. "weak" -> "weakly"**

**Ans:** Thank you. The word "weak" has been changed to "weakly" in the revised version. (P8, line 153)

**Page 8, Line 156. ". . Legates and Willmott (1990). . ."**

**Ans:** Thank you. The expression of the reference has been corrected in the revised version. (P9, line 173-174)

**Page 9**

**Line 176. Please check fonts and alignment for "a", "b", "d".**

**Ans:** Thank you. All formulas and symbols have been rewritten and corrected in the revised version. (P10, line 191-P11, line 205)

**Line 179. Likewise fonts for P, R to match symbols in equation (2).**

**Ans:** Thank you. All formulas and symbols have been rewritten and corrected in the revised version. (P10, line 190-P11, line 204)

**Page 10.**

**Line 190. ". . CSCASS are in Li . ."**

**Ans:** Thank you. The sentence has been revised in the revised version. (P11, line 207)

**Line 193. ". . reproduction of anticyclonic eddies AE1 and AE2 in the NSCS . ."**

**Ans:** Thank you. The section topic has been revised in the revised version. (P11, line 211)

**Lines 197-198. ". . into CSCASS every 3 days . ."**

**Ans:** Thank you. The sentence has been revised in the revised version. (P11, line 214)

**Page 11.**

**Line 209. ". . when the ANP is greater. ."**

**Ans:** Thank you. The sentence has been revised in the revised version. (P12, line 226)

**Lines (209-210). ". . when the ANP is greater than 2 (that is the amplitude greater than 8 cm) . ." These two criteria are not the same thing. It is possible for ANP < 2 but amplitude > 8 cm for a large eddy.**

**Ans:** Thank you. The sentence has been revised in the revised version (P12, line 226). We fully agree with you. But, as our research shown, the ANP > 2 and the amplitude > 8cm is corresponding. As for the existence of the phenomenon you mentioned, more researches are needed to verify it and we will strengthen this research in subsequent investigation.

**Line 210. "well reproduced". Referee 1 main comment 3 asks for an objective measure for good reproduction. Please say how you measure it.**

**Ans:** Thank you. In this study, we can only be qualitative, cannot quantitatively determine the quality of reproduction. As for the criterion, we mainly judge whether the eddy center distance and the amplitude change trend are consistent between the observed and the assimilation results.

**Line 218. ". . CSCASS: the meridional and zonal radii of AE1 . ."**

**Ans:** Thank you. The sentence has been revised in the revised version. (P12, line 235)

**Lines 223-224. ". . (Fig. 7b-7j), e.g. moving northwestward firstly and then southwestward, can generally . ."**

**Ans:** Thank you. The sentence has been revised in the revised version. (P12, line 240-241)

**Page 12.**

**Line 230. ". . their observed amplitude"**

**Ans:** Thank you. The sentence has been revised in the revised version. (P13, line 248-249)

**Line 232. "relatively small, less than 8 cm, . .". But this criterion "8 cm" may depend on altimeter accuracy (Referee 1 main comment 2). You should discuss how the criterion might differ in other contexts. Most readers will not be repeating your work in the NSCS but may want to know a criterion in their context.**

**Ans:** Thank you. The sentence has been revised in the revised version (P13, line 249). Yes, the criterion "8 cm" is relating with the SLA error of assimilation system. For example, the mean SLA error of the CSCASS in the SCS is about 8cm.

**Line 235. "assimilated . ."**

**Ans:** Thank you. The word "assimilating" has been changed to "assimilated" in the revised version. (P13, line 252)

**Page 13.**

**Line 253. "2004 . ."**

**Ans:** Thank you. The year "2003" has been changed to "2004" in the revised version. (P14, line 270)

**Line 255. "disappearance"**

**Ans:** Thank you. The word "disappear" has been changed to "disappearance" in the revised version. (P14, line 272)

**Page 14.**

**Line 273. ". . and then enhancement of AE1 was also predicted . ."**

**Ans:** Thank you. The sentence has been revised in the revised version. (P14, line 290)

**Line 278. Delete "which".**

**Ans:** Thank you. The word "which" has been deleted in the revised version. (P15, line 295)

**Line 279. "continued"**

**Ans:** Thank you. The word "continue" has been changed to "continued" in the revised version. (P15, line 296)

**Line 284. "reproduce . ."**

**Ans:** Thank you. The word "reproduced" has been changed to "reproduce" in the revised version. (P15, line 300)

**Line 287. ". . but the predicted movement is firstly toward . ."**

**Ans:** Thank you. The sentence has been revised in the revised version. (P15, line 304)

**Page 15.**

**Line 296. "slow"**

**Ans:** Thank you. The word "slowly" has been changed to "slow" in the revised version. (P15, line 313)

**Line 309. ". . and movement direction . ."**

**Ans:** Thank you. The word "moving" has been changed to "movement" in the revised version. (P16, line 325)

**Lines 310-311. ". . AE2 are keeping in the consistent trend (Fig. 14e), . .". This is very unclear. Also the figure does not show much consistency between the observed amplitude (overall decrease) and predicted amplitude (overall increase), but the observed and predicted amplitudes are getting closer with time.**

**Ans:** Thank you. The sentence has been revised in the revised version. (P16, line 327-328)

**Page 16.**

**Line 316. "owing to the low amplitude . ."**

**Ans:** Thank you. The sentence has been revised in the revised version. (P16, line 333)

**Line 325. ". . production and predictability . ."**

**Ans:** Thank you. The word "productivity" has been changed to "production" in the revised version. (P17, line 342)

**Lines 327-328. Better ". . The comparisons of AE1 and AE2 observations with CSCASS prediction experiments, which assimilate SLA and SST, show that . ." ?**

**Ans:** Thank you. The sentence has been revised in the revised version. (P17, line 344-345)

**Line 331. "disappearance". Again there is the question of what uncertainty determines the value 8 cm.**

**Ans:** Thank you. As our result shows, the SLA error of assimilation system determines the value 8 cm.

**Lines 333-334. ". . prediction experiments . . with observations . ."**

**Ans:** Thank you. The sentence has been revised in the revised version. (P17, line 350-351)

**Page 17.**

**Line 336. "generation"**

**Ans:** Thank you. The word "generative" has been changed to "generation" in the revised version. (P17, line 353)

**Lines 341-342. ". . reproduction and predictability. As . ."**

**Ans:** Thank you. The sentence has been revised in the revised version. (P18, line 358)

**Lines 348-350. Better "it cannot make up for limitations of numerical model algorithms and resolution. Hence for high-resolution operational oceanography, numerical models . ."**

**Ans:** Thank you. The sentences have been revised in the revised version. (P18, line 364-366)

**Page 20 lines 404-407. These two references should be in reversed order.**

**Ans:** Thank you. The order of the two references has been corrected in the revised version. (P21, line 421-424)

[revised manuscript text omitted]

---

## Author Response (AR3)

Editor, Ocean Science

Email: editorial@copernicus.org

January 23, 2019

Dear Editor:

Thank you for your letter dated January 18 regarding our manuscript entitled "**Could the mesoscale eddies be reproduced and predicted in the northern south China sea: case studies**" (No: os-2018-74), which was submitted to **Ocean Science** for consideration of publication. We have read reviewer's comments carefully. Following your suggestions, we have gone through the manuscript and made the following changes in the manuscript.

The line-by-line replies for reviewer's comments are shown in the attachments. The original comments are quoted in Times New Roman (Bold) and our responses are in Times New Roman.

Once again, thank you and reviewer for the time and energy spending on reading our manuscript and providing constructive comments and suggestions, which are very valuable in improving the quality of our manuscript.

Enclosed, please find our updated manuscript and reply to the reviewer's comments. We hope this version of our manuscript can meet the standard of publication in this journal, and we look forward to receiving further instruction related to this submission.

Sincerely yours

Dazhi Xu

**Comments to the Author:**

**Thank-you for your revisions. I am now asking for some Technical Corrections for clarity (see Details below) and there is one point where I think you still need to answer a point of Referee 1. After this, your manuscript should go direct to the Copernicus production system with no further intervention by me. However, there will be copy editing and I expect that will make changes. You should check that your intended meaning is kept.**

**Ans:** We are very grateful to the editor for carefully reading our manuscript and proposing constructive revisions. We are also very grateful for your efforts and time to improve the quality of this article.

**Details**

**Line 46. "the scales resolved" seems misplaced; it does not fit with the other items in this sentence.**

**Ans:** Thank you. The words "the scales resolved" have been replaced by "the temporal and spatial variability" in the revised version. (P3, line 46)

**Line 57. "marginal sea, in the northwest Pacific, connecting. ." (present wording suggests the Gulf of Mexico is in the northwest Pacific)**

**Ans:** Thank you. The sentence "marginal sea in the northwest Pacific, connecting. ." has been revised in the revised version. (P4, line 57)

**Line 59. "Current (KC), Rossby waves . . SCS and especially the"**

**Ans:** Thank you. The sentence "Current (KC), the Rossby waves and the complex topography, the SCS, especially the" has been revised in the revised version. (P4, line 59)

**Line 64. Delete "in the" at end.**

**Ans:** Thank you. The words "in the" have been deleted in the revised version. (P4, line 64)

**Line 86. Delete "its".**

**Ans:** Thank you. The word "its" has been deleted in the revised version. (P5, line 86)

**Line 97. Delete "appear"**

**Ans:** Thank you. The word "appear" has been deleted in the revised version. (P6, line

97)

**Line 98. "show that an ocean model including tides or"**

**Ans:** Thank you. The sentence "show that the ocean model includes tides or" has been revised in the revised version. (P6, line 98)

**Line 101. Delete "about".**

**Ans:** Thank you. The word "about" has been deleted in the revised version. (P6, line 101)

**Lines 102-104. ". . two typical NSCS anticyclonic eddies (Wang et al., 2008), chosen as representing different generation mechanisms and surviving long enough to be useful, with focus . ."**

**Ans:** Thank you. The sentences "..two typical anticyclonic eddies (Wang et al., 2008), owing to be represented different generation mechanisms and survive long enough to be useful, in the NSCS with focus.." have been revised in the revised version. (P6, lines 102-104)

**Line 107. "trajectories"**

**Ans:** Thank you. The word "trajectory" has been replaced by "trajectories" in the revised version. (P6, line 107)

**Lines 111-112. "In this study, altimetric data in 2003-2004 was selected, including along-track SLA, totalling 29 passes (about 9300 points) over the domain of CSCS. Considering . ."**

**Ans:** Thank you. The sentences "In this study, the altimetric data between 2003-2004, which includes along-track SLA, totally 29 passes (about 9300 points) over the domain of CSCS was selected. Considering.." have been revised in the revised version. (P6, lines 110-111)

**Line 119. Delete "thus"**

**Ans:** Thank you. The word "thus" has been deleted in the revised version. (P7, line 118)

**Line 121. "(2012), was . ."**

**Ans:** Thank you. The sentence "..Xu et al. (2012) was used.." has been revised in the revised version. (P7, line 119)

**Line 180. "climate" -> "climatology"**

**Ans:** Thank you. The word "climate" has been replaced by "climatology" in the revised version. (P9, line 179)

**Lines 226-227. "fig. 5 shows . . AE2 can be well reproduced". Actually figure 5 shows ANP and the model result is not close to the observed ANP. Referee 1 main question 3 was how do you measure "well reproduced"? Perhaps your answer is in the next paragraph lines 228-241? Please correct and clarify this for readers.**

**Ans:** Thank you. As the editor pointed out, the answer to "the referee 1 main question 3" is indeed in paragraph 2 of page 12 (lines 230-241). And this paragraph (P11-12, lines 222-226) is the answer for referee 2 question 4. Furthermore, in order to eliminate the reader's misunderstanding, we deleted word "well" in line 225 (P11) in the revised version.

**Line 227. ". . greater than 8 cm in our runs) . .". ANP > 2 and amplitude greater than 8 cm may correspond in your example(s) but this is not a universal relation.**

**Ans:** Thank you. The sentence ".. greater than 8 cm).." has been revised in the revised version. (P11, line 225)

**Lines 228-229. "Besides, we also use independent evaluation. Fig.6 compares the assimilating results of AE1 with observations . ."**

**Ans:** Thank you. The sentences "Besides, we also use the independent evaluation, Fig.6 compared the assimilating results of AE1 with the observations.." have been revised in the revised version. (P12, lines 227-228)

**Line 232. "CSCASS; the pink curves"**

**Ans:** Thank you. The sentence "..CSCASS, with the pink curves.." has been revised in the revised version. (P12, line 231)

**Line 248. "amplitude is greater . . "**

**Ans:** Thank you. The sentence ".. amplitude greater.." has been revised in the revised version. (P12, line 247)

**Line 270. ". . Exp5 is set up . ."**

**Ans:** Thank you. The sentence ".. Exp5, is setting up.." has been revised in the revised version. (P13, line 269)

**Line 290. ". . AE1 was also predicted . ."**

**Ans:** Thank you. The sentence ".. AE1 can also been predicted.." has been revised in the revised version. (P14, line 289)

**Line 310. Delete "thus"**

**Ans:** Thank you. The word "thus" has been deleted in the revised version. (P15, line 309)

**Line 315. ". . centres of the two almost coinciding. The . ."**

**Ans:** Thank you. The sentence ".. center of the two almost coincide. The.." has been revised in the revised version. (P16, line 314)

**Line 326. "moving" -> "movement"**

**Ans:** Thank you. The word "moving" has been replaced by "movement" in the revised version. (P16, line 325)

**Lines 334-336. ". . The observed amplitude of AE2 decays continually at this stage, but the predicted amplitude is almost constant. In . ."**

**Ans:** Thank you. The sentences ".. The amplitude of AE2 from the observation decays continually at this stage, but the amplitude of the predicted almost keeps constant. In.." have been revised in the revised version. (P16, lines 333-335)

**Figure 5 caption line 550. "observed" -> "observations".**

**Ans:** Thank you. The word "observed" has been replaced by "observations" in the revised version. (P27, line 540)

**Figure 6 caption line 554. Delete "of".**

**Ans:** Thank you. The word "of" has been deleted in the revised version. (P28, line 543)

**Figure 7 line 562. ". . figure 6, . ."**

**Ans:** Thank you. The figure number "4" has been corrected and replaced by "6" in the revised version. (P29, line 551)

**Figure 8 caption line 583. Delete "moving"**

**Ans:** Thank you. The word "moving" has been deleted in the revised version. (P32, line 564)